# Contact with Nature in Social Deprivation during COVID-19: The Positive Impact on Anxiety

**DOI:** 10.3390/ijerph20146361

**Published:** 2023-07-14

**Authors:** Ferdinando Fornara, Elena Rinallo, Massimiliano Scopelliti

**Affiliations:** 1Department of Education, Psychology, Philosophy, University of Cagliari, 09123 Cagliari, Italy; ffornara@unica.it; 2Department of Human Studies, Libera Università Maria Ss. Assunta (LUMSA University), 00193 Rome, Italy; m.scopelliti@lumsa.it

**Keywords:** anxiety, connectedness to nature, contact with nature, COVID-19, green spaces, restorativeness, well-being

## Abstract

The 2019 outbreak of the novel coronavirus disease (COVID-19) had a devastating impact on millions of people worldwide. Following the constantly changing course of the pandemic, the Italian government massively restricted public and private life to prevent the further spread of the virus. Unfortunately, lockdown policies negatively impacted many people’s mental and physical health. Numerous studies recognized an essential role of urban green areas in promoting human well-being. The present study aims to evaluate the effect of personal dispositions towards nature, measured using the connectedness to nature scale (CNS) and actual contact with green spaces (CwN) on human well-being (i.e., anxiety) and medicine intake during COVID-19 lockdowns. A total of 637 Italian residents answered a survey aimed at gathering information about the above variables. A series of path analyses were performed. The results showed that the CNS was positively associated with the CwN, and the latter, in turn, was negatively associated with anxiety. Finally, anxiety was positively related to medicine intake. In sum, these results identify the positive role of person–nature relationships for individual well-being during COVID-19 restrictions.

## 1. Introduction

As of 19 April 2023, the WHO flagged 6,908,554 deaths in the world due to the novel coronavirus disease (COVID-19). As of the same date, there have been 763,740,140 confirmed cases in 219 countries and territories. As for the specific situation in Italy, 25,737,170 confirmed cases and 185,993 deaths were reported during the same period [1]. Since the early stages of the COVID-19 outbreak, the National Healthcare Service, which provides universal access to care, faced an ever-increasing pressure, which almost led to its collapse [2]. With the healthcare system in difficulty, numerous healthcare services (e.g., surgical services) were delivered with an extreme delay [3]. In addition, post-traumatic stress disorder (PTSD) [4], anxiety, insomnia, and depression diagnoses increased among healthcare workers [5]. 

With the aim of easing the pressure on the National Health System and limiting the spread of the virus, the Italian Government imposed a national lockdown on 11 March 2020, and then gradually resumed suspended economic and social activities starting 4 May of the same year. From March to May, Italian citizens were restricted from leaving their homes, except only under specific circumstances. These circumstances included medical needs, grocery or pharmacy shopping, and commuting to essential workplaces, while all other activities were either suspended or transitioned to remote work (i.e., smart working) [6]. Citizens were also allowed to do physical activities in green areas within the municipality [6].

Even though lockdown policies helped contain the spread of the virus, they negatively impacted many people’s mental and physical health globally [7]. Home confinement substantially impacted weight-related behaviors, including healthy eating and physical activity, hindering the implementation of healthy weight management behaviors in both children and adults [8], with the latter presenting an increased cardiovascular risk burden [9]. 

Lockdown should be considered a multi-faceted experience [10]. Depression and anxiety affected thousands of people, causing functional impairments [11]. The presence of the same symptoms in parents as well appeared to be associated with higher parental perceived stress and, in turn, with potential child abuse [12]. Pregnancies were heavily affected by COVID-19-related home confinement as well. The most severe anxiety and depression symptoms regarded concerns about the threat of COVID-19 to both the life of the mother and the baby, as well as concerns about social isolation, not getting the necessary prenatal care, and the possible ensuing tensions in relationships [13]. For their part, children and adolescents began to manifest red flagged behaviors for emotional distress, such as physical distancing and decreased activity, which, instead were considered adaptive for these age groups [14]. Finally, evidence about the psychological impact of COVID-19 pandemic pointed to incremental substance use and self-medication as a means of coping [15]. Prescription drugs were included among the most commonly abused substances [15,16].

### 1.1. Personal Disposition towards Nature and Contact with Nature for Human Well-Being

The importance of nature for human well-being has been widely debated in scientific literature. Since Wilson’s seminal work on the biophilia hypothesis [17], the idea of an innate disposition for human beings to positively respond to natural environments as a result of evolution has been recognized and empirically investigated. In this regard, several constructs have been proposed. Recently, Tam [18] empirically examined a broad range of these concepts and measures, with the goal of clarifying the similarities and differences between them. His findings suggested that they could be viewed as indicators of the same underlying construct. Nonetheless, the findings also indicated the presence of subtle divergences that warrant recognition. In the following section, a brief description of several constructs conceptualizing the connection to nature is provided.

### 1.2. Exploring the Evolution of the Connection to Nature Conceptualizations: An Overview

Mayer and Frantz [19] theoretically developed the construct of connectedness to nature, referring to the individuals’ experiential sense of oneness with the natural world, and then validated a scale for its measurement (CNS). This tool has been widely employed in the literature, and a consistent association with human well-being has been reported. For example, across five studies, Cervinka et al. [20] found a significant association between the CNS and a series of measures of well-being, including mood, life satisfaction, and both physical and psychological well-being. Howell et al. [21] found a positive relationship between the CNS and measures of emotional, psychological, and social well-being. Moreover, a significant association between the CNS and mindfulness emerged. In a study on contact with nature in urban parks in Bogotà, Scopelliti et al. [22] reported a significant relationship between the CNS and physical and psychological well-being, noting a stronger association for low-income residents. More focused research has identified fine-grained mechanisms through which a connectedness to nature may affect human well-being. To give some examples, Mayer et al. [23] reported a positive effect of nature exposure on the ability to reflect on life’s problems and positive effects, where the increase in a connectedness to nature was a key process for mediating those beneficial outcomes. In a cross-cultural multi-study, Capaldi et al. [24] found that engagement with natural beauty promotes well-being through the indirect effect of connectedness to nature. In Liu et al.’s study [25], the awe of nature, a positive emotion arising from the perceived vastness of the natural environments, improved the participants’ well-being by increasing their connectedness to nature. Similar findings have emerged with reference to clinical populations. For example, Keenan et al. [26] evaluated the effectiveness of nature-based interventions for patients with anxiety and depression, taking into account the role of the CNS in a pre–post experimental design. This study showed that the increase in the levels of hedonic (positive emotion) and eudemonic (self-awareness) well-being over time was explained by changes in the CNS.

Within a more comprehensive framework, Nisbet et al. [27] proposed the concept of nature relatedness, describing individual levels of connectedness with the natural world, and encompassing the cognitive, affective, and experiential aspects of this relationship. Through two studies, the authors developed a valid and reliable scale to measure this trait, which was conceived to be “relatively stable over time and across situations” (p. 718). Nature relatedness was found to be associated with time spent in nature, environmental concern, and pro-environmental behavior. A short version of the nature relatedness scale was also proposed and empirically tested [28]. The role of nature relatedness in human well-being has also empirically emerged. Nisbet et al. [29] found a significant association between nature relatedness and several measures of well-being across different populations. In addition, the authors identified a mediating role of nature relatedness in the relationship between environmental education and a further measure of well-being (i.e., vitality). Zelensky and Nisbet [30] also outlined a significant association between nature relatedness and happiness, over and above the role of other measures of connection, including one’s friends or country. Martyn and Brymer [31] reported a significant association between nature relatedness and lower levels of anxiety. Similar findings emerged in Lawton et al.’s study [32]. In a further study by Dean et al. [33] involving a large and stratified Australian sample, distinct health effects of nature relatedness emerged. In particular, the aspects of nature relatedness reflecting an enjoyment of nature were significantly associated with reduced ill health, while the aspects of nature relatedness reflecting self-identification with nature were associated with increased depression, anxiety, or stress, after accounting for potential confounding factors. More recently, Aruta and Pakingan [34] tested the validity of nature relatedness in a different cultural context (i.e., Philippines) and found a positive association with green purchase intentions. 

In a different vein, Perkins [35] proposed and empirically validated the construct of love and care for nature (LCN), stressing the emotional aspects in people’s relationships with nature based on a clear recognition of its intrinsic value and a spiritual dimension, and associated with feelings of care and responsibility for its protection. This tool urged the importance of focusing on non-cognitive aspects of human–nature relationships, as recognized by Perrin and Benassi [36] in their five-study critical analysis of the CNS. Using the LCN scale, Wu and Zhu [37] investigated the importance of this emotional disposition within a well-established model for predicting pro-environmental behavior (PEB), namely the value-belief-norm theory (VBN) [38]. VBN states that personal values, environmental worldviews, beliefs, and personal norms are associated in a sequential chain leading to PEBs. The predictive power of VBN in explaining PEBs has been consistently recognized in the literature across social groups and cultural contexts [39,40,41,42,43,44]. In their study, Wu and Zhu [37] identified a key role for LCN within the VBN theory. In particular, a significant association between LCN and both biospheric values and personal norms emerged within the causal chain of the model explaining low-cost (i.e., easy and convenient) pro-environmental behavior. Moreover, LCN showed a direct effect on high-cost (i.e., difficult and inconvenient) pro-environmental behavior. More recently, in a pre–post experimental study based on a nature excursion with a contrast group, Garza-Teran et al. [45] reported an increase in the positive effects and several measures of human–nature connections, including LCN, after the excursion and a positive association between LCN and satisfaction with life.

Based on the concept of empathy, Tam [46] conceived the construct of dispositional empathy with nature (DEN), referring to a personal tendency to understand and share the emotional experience of the natural world. Through five studies, the author tested a scale for the measurement of DEN, and empirically distinguished this construct from both empathy between human beings and other well-established variables associated with pro-environmental behavior, including personality traits, values, environmental concerns, and emotional involvement with nature. Empathy for nature, and DEN in particular, have begun to emerge as relevant psychological constructs for environmental research. For example, Scopelliti et al. [47] have found a significant association between DEN and PEBs, with activism among PEBs showing the most significant association. Similarly, Young et al. [48] convincingly discussed that empathy for animals may promote pro-environmental behavior changes, and Williams et al. [49] recently developed a review of studies showing a significant association between the anthropomorphism of nature and several pro-environmental variables.

Overall, a highly significant meta-analysis based on 30 samples and more than 8000 respondents showed a significant association between a connection with nature measured using different tools and several measures of well-being, including vitality, positive effects, and satisfaction with life [50]. Similarly, a meta-analysis conducted by Pritchard et al. [51], including 20 samples and more than 4000 respondents, found a significant association between nature connectedness and eudaimonic well-being. These studies further stressed the importance of considering the role of personal dispositions when analyzing the positive impacts of contact with nature for human well-being.

Beyond a personal disposition towards nature, contact with nature in itself has been consistently found to promote health and well-being [52,53,54], especially with reference to specific populations, such as adolescents [55]. Several mechanisms have been identified explaining these beneficial outcomes, including the improvement in air quality, increase in physical activity, facilitation of social interactions and cohesion, and psychological restoration, referring to the reduction in stress and mental fatigue promoted by natural environments [52,56]. Psychological restoration has been framed within two main evolutionary theories, namely the Stress Reduction Theory (SRT) [57] and the Attention Restoration Theory (ART) [58,59], which are both supported by compelling empirical evidence [60,61,62]. ART also states that psychological restoration is associated with the perception of some environmental features, whose levels can be considered a measure of the restorative potential of environments. Several tools have been developed to measure the perceived restorativeness of environments, with the Perceived Restorativeness Scale (PRS) [63] being the most used in the literature.

Contact with nature was shown to have positive effects on human health and well-being during the COVID-19 pandemic. In a study in the Netherlands, Shentova et al. [64] investigated the positive effects of nearby nature on well-being. Their results showed that the quantity of the greenery in the residential area was a relevant factor, but the quality was more strongly associated with well-being. In particular, well-maintained, attractive, and varied streetscape greenery were key aspects. In their study on the effects of lockdown severity on well-being across nine countries, Pouso et al. [65] found that nature views from homes and access to outdoor spaces were significantly associated with lower symptoms of depression and anxiety, also taking into account the role of several socio-demographic variables as covariates. This result was even stronger for people under strict lockdown. A positive effect of the frequency of greenspace use and the view of nature from home windows on depression and anxiety also emerged in a study on the Japanese population [66] and among U.S. residents in Denver, where contact with nature was measured through objective aerial imagery of nature nearby the participants’ residences and the perception of the amount, quality, and access of green spaces [67]. In a 20-month follow up study on approx. 20,000 U.K. residents, the percentage of green cover around the home was still significantly associated with lower anxiety symptoms [68]. Egerer et al. [69] reported a strong importance of gardening for stress reduction during the first wave of the pandemic in their data collection across several countries. Similar findings in terms of anxiety reduction associated with gardening among U.S. adults emerged in Gerdes et al.’s study [70].

Based on the above literature, the present study was aimed at gaining a better understanding of the mechanisms through which a personal disposition towards nature and contact with nature may have promoted well-being in terms of anxiety and medicine intake reduction during the COVID-19 pandemic. 

The research hypotheses were based on a sequential model which includes the following assumptions.

**H1.** 
*Individuals who are more connected to nature are more likely to seek out and have contact with nature.*


**H2.** 
*The greater the contact with nature, the higher the degree of perceived restorativeness.*


**H3.** 
*The higher the degree of perceived restorativeness, the lower the level of anxiety.*


**H4.** 
*Individuals who experience less anxiety exhibited a lower utilization of medications for psychological well-being during the pandemic, in comparison to the period before the COVID-19 emergency (i.e., medicine intake. Please note that in our study, “medicine” refers specifically to medications taken for psychological well-being, excluding illegal substances, alcohol, and nicotine.). Both indirect (mediated) paths and direct paths between non-proximal variables within the sequence were also tested.*


## 2. Materials and Methods

### 2.1. Sampling Procedure

A total of 637 individuals residing in various cities and towns across Italy took part in the study. Due to the COVID-19 pandemic and related restrictions, the participant recruitment was constrained. To overcome this challenge, we employed a retrospective analysis with a snowball sampling technique, where a small initial group of participants contacted others to expand the sample when the restrictions were less stringent in February–April 2022. 

Near-graduate students received face-to-face training to recruit participants using a variety of channels, such as relatives, close friends, and social networks. These students actively facilitated the engagement of potential participants, guiding them through the process of completing an online questionnaire after securing informed consent. The primary aim of this methodological approach was to ensure the acquisition of a diverse—and as representative as possible—sample of participants.

### 2.2. Measures

An online questionnaire was employed to assess the constructs of interest. The questionnaire comprised the following measures, each utilizing a seven-point Likert-type response scale, where the participants were required to indicate their level of agreement or disagreement. The Appendix A include a copy of the administered questionnaire, available for reference.

A connectedness to nature scale [19] consisting of 15 items (e.g., “I think of nature as something I belong to”) was utilized to capture the individuals’ subjective sense of belonging to nature. The Italian translation used by Scopelliti et al. [47] was employed. This scale underwent a meticulous selection process, resulting in the removal of three items to enhance its psychometric properties. The scale demonstrated a high level of internal consistency, as indicated by a Cronbach’s alpha coefficient of 0.89. Contact with nature was used to evaluate the individuals’ engagement with the natural environment, a specifically designed ad hoc measure comprising five items was administered. This measure aimed to gauge the frequency of the individuals’ involvement in nature-related activities, reflecting their contact with natural settings (e.g., “I have spent time in nature”). The internal consistency analysis indicated a very good level of reliability, with a Cronbach’s alpha coefficient of 0.88.The Perceived Restorativeness Scale [63,71], consisting of eight items, explored the participants’ subjective evaluation of the extent to which nature provides restoration (e.g., “There is a lot to explore and discover”). The scale demonstrated an very good internal consistency, with a Cronbach’s alpha coefficient of 0.87.The Anxiety STAI–State Short [72] scale comprised six items, capturing the intensity of anxiety symptoms experienced by individuals (e.g., “I felt worried”). The measure exhibited a good internal consistency, as evidenced by a Cronbach’s alpha coefficient of 0.84. Medicine intake, a single-item ad hoc measure, was employed to evaluate the participants’ medicine intake for psychological well-being in comparison to the period preceding the COVID-19 emergency (i.e., “I have taken more medicines for my psychological well-being than before the COVID-19 emergency”). This item served as a concise indicator of the changes in medication usage specifically related to psychological concerns.

Socio-demographic information was also collected, including age, gender, education level, job activity, and place of residence.

### 2.3. Statystical Analysis

The collected data were analyzed using the Jamovi, 2.3.21.0 version statistical software [73]. The significance threshold was set to *p* < 0.05. A Pearson’s r correlation analysis was used to evaluate the associations between the observed variables. The research hypotheses were verified by means of a path analysis performed using the PATHj Directory, 0.8.0 version [74], implemented in the free and open statistical software Jamovi, 2.3.21.0 version [73]. An adjusted bias-corrected bootstrap procedure (N = 1000) was employed to assess the indirect effects (i.e., αβ), their standard errors (i.e., S.E.), and 95% confidence intervals (i.e., 95%C.I. [LL, UL]) [75].

## 3. Results

### 3.1. Sample Characteristics

A total of 637 participants responded to the online questionnaire, providing valuable data for the study. The age range of the participants varied from 15 to 88 years, with a mean age of 34.5 years (SD = 15.3), thereby guaranteeing a comprehensive representation of various age groups. The gender distribution revealed that 72.2% of the respondents were women. Notably, individuals between the ages of 21 and 30 constituted the largest subgroup, accounting for 50.3% (320 individuals) of the respondents. In terms of educational attainment, a significant proportion of the respondents (7%, 353 individuals) held a master’s degree. Additionally, the participants identified themselves mainly as students (33.6%, 214 individuals) or employees (20.3%, 129 individuals). For a comprehensive overview of the participants’ characteristics, refer to Table 1.

### 3.2. Descriptive Statistics

Table 2 shows the descriptive statistics concerning the variables being studied and the bivariate correlations between them. The log10 transformation of the medicine intake was used for the analysis in order to rely on a more normal-like distribution of the outcome variable.

The preliminary analyses showed no significant difference between the males and females with reference to their contact with nature (F_(1, 628)_ = 1.38, n.s), connectedness with nature (F_(1, 628)_ = 2.18, n.s), or perceived restorativeness (F_(1, 628)_ = 2.93 n.s). Conversely, the females reported higher levels of anxiety (F_(1, 628)_ = 37.16, *p* < 0.000) and medicine intake (F_(1, 628)_ = 3.97, *p* = 0.047).

### 3.3. Path Analysis

The tested path analysis included all the hypothesized direct and indirect paths, including the relationships between the non-proximal variables. Gender and age were inserted in the model as covariates.

Figure 1 shows the results of the final path analysis model. Despite a significant positive relationship between contact with nature and the perceived restorativeness that emerged in the model, thus confirming H2, we decided to exclude the perceived restorativeness from the final model since its hypothesized (negative) association with anxiety was not significant, thus disconfirming H3. We also tested the possible role of perceived restorativeness as a moderator between contact with nature and anxiety, but again the result was not significant. 

Nevertheless, the tested relationship between the supposed antecedent of perceived restorativeness, i.e., contact with nature, and (low) anxiety (H3 new) was found to be significant. Hence, a shorter sequential chain was represented in the model. The model fit indices were acceptable where χ^2^(3) = 5.71; *p* = 0.127; RMSEA = 0.038, with a 95% confidence interval for RMSEA = 0.000–0.085; the test of close fit RMSEA < 0.05: *p* = ns; SRMR = 0.019; TLI = 0.91; and CFI = 0.98. Consistently with the hypothesized sequential chain, a higher connectedness to nature (H1) was associated with a more frequent experience of contact with nature (B = 0.371, SE = 0.074; 95% CI [0.216, 0.511], β =.179, *p* < 0.001). This was related to a lower level of anxiety (H3new) (B = −0.113, SE = 0.031; 95% CI [−0.054, −0.176], β = −0.145, *p* < 0.001), which in turn facilitated (H4) the reduction in medicines intake (B = 0.061, SE = 0.009; 95% CI [0.077 0.041], β = 0.278, *p* < 0.001).

The direct paths between the non-proximal variables were all not significant, whereas the indirect effect of the 1^st^ level variable (i.e., connectedness to nature) on the outcome variable (i.e., medicine intake) through the sequential mediation of contact with nature and anxiety was significant (B = −0.003, SE = 0.001; 95% CI [−0.005 −0.001], β = −0.007, *p* = 0.005). Thus, a full mediation model emerged. Analogously, the indirect effect of contact with nature on the medicine intake appeared was fully mediated by anxiety (B= −0.007, SE = 0.002; 95% CI [−0.012 −0.003], β = −0.040, *p* < 0.001). Additionally, the covariate age was found to have a significant negative effect on contact with nature (β = −0.134, *p* = 0.001) and anxiety (β = -.105, *p* = 0.007), and a significant positive effect on the medicine intake (β = 0.121, *p* = 0.002) after controlling for the effects of the independent variable. In other words, as the age increased, medicine intake increased, while contact with nature and anxiety decreased. Finally, females showed significantly more anxiety than males (β = −0.226, *p* < 0.001).

## 4. Discussion

This study helped shed light on some key mechanisms referring to human–nature relationships leading to well-being during the COVID-19 pandemic. In particular, a sequential chain was tested, including the participants’ personal disposition to feel related to the natural environment (namely, connectedness to nature [18]), their actual contact with nature, and their perception of its restorative potential [51,56,58] as the antecedents of well-being outcomes in terms of (a lower level of) anxiety and, ultimately, medicine intake. Both anxiety and medicine intake were identified as negative outcomes of the COVID-19 pandemic and related lockdowns worldwide [7,10,11,13,14,15,16].

The expected positive relationship between connectedness to nature and actual contact with (i.e., experience of) nature was significant, thus confirming H1. This was consistent with the literature on contact with nature, that widely discussed the antecedents and consequences in terms of attitudes, emotions, behaviors, and overall effects on health and well-being [52,76,77]. In particular, a positive relationship between nature connectedness and nature contact was largely recognized [26,78]. Although past research mainly investigated how contact with nature may lead to a stronger connectedness, this relationship is likely to be bidirectional. Especially when we try to understand what leads people to select natural environments as places for experiencing well-being, the reverse relationship may occur, as shown in our results. In fact, it is important to recall that the focus of this study was the period of national lockdown, when citizens’ access to green areas was limited. Thus, it is likely that those people feeling more connected with nature were more motivated to search for green areas where they could spend time. A complementary phenomenon that should be tested in future research supposes that more connected people would experience lower levels of well-being due to the limitations of moving and spending time in green areas.

In line with the literature [51,56,58], as hypothesized, contact with nature was found to be associated with a higher perception of its restorative potential (H2). However, contrary to our hypothesis (H3), it was not significantly related to a lower level of anxiety. On the other hand, a direct significant link emerged between contact with nature and a reduction in anxiety, thus corroborating the substituted hypothesis (H3new) formulated for the continuity of the sequential chain in the tested model. In fact, the relationship between nature contact and the reduction in anxiety and its symptoms was consistently found in the literature, with reference to both clinical and non-clinical populations [79,80,81]. Finally, individuals feeling less anxious about the pandemic situation showed a lower degree of medicine intake, as expected (H4). The lack of an association between the perception of the restorative potential of nature and well-being was apparently contrasted with the well-established empirical literature on the relationships between restorative natural environments and anxiety reduction, along with reference to recent findings on nature experiences during the COVID-19 pandemic [82,83]. However, some possible explanations can be proposed. First, those studies mainly measured anxiety as an outcome of the experience of natural environments, which was assumed to be restorative without measuring the cognitive perception of their restorative properties by the respondents. Second—and related to the previous point—nearby nature that was accessible by residents during the COVID-19 pandemic may have suffered from a lack of maintenance, while nature contact investigated by other studies often used an experimental approach through videos or the simulation of high-quality natural scenes. This may have hindered the role of cognitive restoration in promoting well-being. Third, and more important, this apparently unexpected finding was compatible with the discussion proposed by Kaplan [59] about the potential mechanisms through which affective and cognitive restoration may interact, which are inherently different processes. According to SRT [57], stress reduction occurs as an immediate affective response to natural environments due to some gross aspects (e.g., vegetation, visual depth) evolutionarily promoting pleasure and a reduction in arousal. Our result about the relationship between mere nature contact and anxiety reduction was in line with this assumption. Conversely, ART [58] states that the cognitive process of restoration needs more interaction time with nature—which was not always allowed under lockdown restrictions—and the recovery of mental fatigue is the key outcome. However, further research is undoubtedly needed to better understand these fine-grained mechanisms that are often neglected in the literature, which has mainly focused on the measure of stress reduction and cognitive improvement as simultaneous outcomes [84,85,86]. With reference to the positive relationship between the levels of anxiety about the pandemic situation and medicine intake, the role of other factors beyond contact with nature, which were not considered in this study, need to be acknowledged. To provide an example, the recent literature on the COVID-19 outbreak consistently identified the importance of social support to reduce anxiety levels [87,88].

The study’s findings also revealed that age had a significant impact on contact with nature, anxiety, and medicine intake. Specifically, the negative relationship between age and both contact with nature and anxiety suggested that older individuals engaged less in nature-related activities and experienced lower anxiety levels. This was especially relevant given recent research suggesting that during the COVID-19 pandemic adults spent less time outside due to pandemic-related restrictions, which has been linked to poorer mental health outcomes [89]. Importantly, younger adults were found to experience higher levels of anxiety during the pandemic than older adults, which was coherent with previous studies. The lower anxiety levels in older adults may have been due to the result showing a positive relationship between age and medicine intake, with no significant association with the CNS, indicating that as people age, they may rely more on medication to manage their health issues [90]. Furthermore, the study highlights a significant gender difference in anxiety levels, with females reporting significantly higher anxiety levels than males. This finding was consistent with the previous research [91,92] and can be explained by a variety of factors referring to biology, personality, stress and coping strategies, and socialization (see [93] for a comprehensive review).

In sum, the tested model showed a sequential chain with connectedness to nature positively associated with contact with nature. This in turn showed a negative relationship with the level of anxiety. Consequently, lower anxiety levels were associated with a decrease in medicine intake.

## 5. Limitations 

This study had several limitations that should be acknowledged. First, the data were obtained using a cross-sectional design, thus making it difficult to identify clear causal relationships between the variables we investigated. Second, the retrospective procedure for the data collection may have partly biased our findings. However, past research showed the validity of the retrospective analyses [94], even in the case of adults reporting information about their children [95]. Third, our sample was not representative of the general population, and different results might emerge through a more adequate sampling procedure. Fourth, the participants were recruited using a snowball procedure. This sampling method reduced the possibility to generalize our findings to the whole population and to test more complex models. Additionally, it was likely that more motivated youth or those interested in environmental issues decided to participate. To overcome this limitation, during the data collection, we proposed to reach a certain degree of sample diversity by beginning the sample within data collecting contexts that were as diverse as possible. One noteworthy limitation of this study pertained to the potential influence of participants’ residential areas and occupations on their contact with nature. Individuals in rural areas were more likely to encounter ample opportunities for contact with nature compared to their urban counterparts. Moreover, individuals engaged in occupations such as farming, fishing, or forestry were inherently exposed to regular and direct interactions with the natural environment, which was in contrast to individuals working in office-based professions. Nonetheless, these findings outlined an interesting mechanism through which the personal dispositions towards nature and actual experiences with nature can concur to promote well-being.

## 6. Conclusions and Future Prospects

This study investigated the relationships between human dispositions towards nature, nature contact, and well-being during the COVID-19 pandemic, identifying a positive association between nature connectedness, nature contact, and their beneficial outcomes. These results could be relevant for practical implications. Basically, the study stressed the relevant role of contact with nature for well-being during emergencies that impose restrictions on freedom of movement. This can be considered by the authorities for the huge social and economic impact of a strict confinement of the population. It would also be important for public communication to emphasize the beneficial effects of nature contact among the general public in order to promote healthier lifestyles. In addition, the study’s findings suggest that nature contact and connectedness may serve as protective factors against the negative mental health impacts of the COVID-19 pandemic, such as anxiety and depression. This highlights the potential value of incorporating nature-based interventions into mental health care and support services during times of crisis.

Furthermore, the study’s results emphasize the need for increased access to natural spaces, particularly in urban areas where green spaces may be limited. This could include initiatives such as community gardens, urban parks, and green infrastructure projects that prioritize the creation and maintenance of natural environments. Especially for people with limited mobility, contact with nature can be a feasible way to promote well-being, even through the use of virtual reality, as recently shown by Sanchez-Nieto et al. [96].

## Figures and Tables

**Figure 1 ijerph-20-06361-f001:**
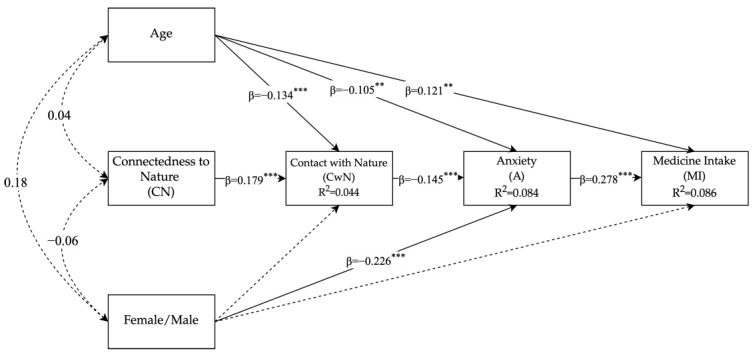
Path diagram model of the association between CN, CwN, anxiety, and medicine intake. Please note: ** *p* < 0.01; *** *p* < 0.001. The path analysis shows the associations between CN, CwN, anxiety, and medicine intake, controlled for age and gender. The coefficients presented are the standardized linear regression coefficients.

**Table 1 ijerph-20-06361-t001:** Socio-demographic characteristics of participants (N = 637).

Demographic Variable	Category	Frequency	Percentage (%)
Gender	Male	175	27.8%
	Female	455	72.2%
Age	18–20	52	8.2%
	21–30	320	50.3%
	31–40	78	12.3%
	41–50	44	6.9%
	51–60	82	12.9%
	Over 60	60	9.4%
Education Level	Junior High school	35	5.5%
	High School	317	49.8%
	Bachelor	111	17.4%
	Master	140	22.0%
	Ph.D.	34	5.3%
Occupation	Students	214	33.6%
	Employees	129	20.3%
	Other Profession	294	46.1%

**Table 2 ijerph-20-06361-t002:** Descriptive statistics and bivariate correlations between the variables being studied (S2).

Variable	N	M	SD	1	2	3	4	5	6
1. Medicine Intake	637	0.18	0.28	1					
2. Anxiety	637	2.33	1.27	−0.265 ***	1				
3. Perceived Restorativeness	637	4.31	1.12	−0.021	0.032	1			
4. Contact with Nature	637	3.07	1.63	0.015	0.140 ***	0.336 ***	1		
5. Connectedness to Nature	637	3.90	0.80	0.054	−0.040	0.502 ***	0.155 ***	1	
6. Age	637	34.55	15.34	0.084 *	−0.105 **	0.039	−0.138 **	0.072	1

Please, note: * *p* < 0.05; ** *p* < 0.01; *** *p* < 0.001.

## Data Availability

The data that support the findings of this study are available from the corresponding author, [E.R.], upon request.

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
