# Peer review of "Contact with Nature in Social Deprivation during COVID-19: The Positive Impact on Anxiety"

_ijerph, 2023, doi:10.3390/ijerph20146361_

Round 1
Reviewer 1 Report
This is an interesting study that brings together contact and connectedness to nature, and its relationship with anxiety and medicine intake. It was a pleasure to read and I would like to see it published. However, I would recommend the following changes are made to make the paper even stronger.
General language/English needs to be looked at, for precision and meaning (e.g. smart working - does that refer to working from home?)
Tenses need to be looked at. For example: “There are currently 763.740.140confirmed cases in 219 countries and territories.” It seems that this should read ‘There have been 763,740,140 confirmed cases [etc] to date.’
Line 19: Path analyses? Should be plural, I think.
Line 21: Medicine - does this refer to prescription drugs/medication (as in line 63)? If so, I recommend that the same term is used throughout the article to refer to medication.
The authors go through various related constructs (Connectedness to Nature, Nature Relatedness etc). A useful reference would be the Tam (2013) review of the constructs and how they relate.
Line 156: The Capaldi et al meta-analysis cannot be described as ‘recent’ - it is now almost ten years old.
They hypotheses need to be rephrased, as the syntax is incorrect.
H4 needs some clarification as to what the term ‘medicine’ refers to. Prescription medication? Only medication for mental/psychological health, or all medication? Does it include illegal substances or other drugs, such as alcohol and nicotine?
Can you please talk a little about the sampling method? Was it a convenience sample, or possibly a snowball sampling technique. Why do you think there was
I would like some information on the CNS in Italian (I assume it was administered in Italian). How was it translated, or was there a validated translation used? Was the translated scale validated in the population or was it assumed that it would capture the same construct as the English language scale?
In the measures, as described, is the Cronbach’s alpha the one from this study? How does it compare to the validation/reference study?
Line 344: “In sum, the tested model has shown a sequential chain, with connectedness to nature positively associated with contact with nature, and this in turn showing a negative relationship with the level of anxiety, which is finally positively associated with medicine intake.” Should this read negatively associated with medicine intake? As in, the more anxiety the less medicine the person takes?
As above - some changes needed for clarity and precision.
Reviewer 2 Report
Thank you for allowing me to review the manuscript. The manuscript deals with a valuable area of the relationship between contact with nature and its impact on anxiety during the COVID-19 pandemic. Although the research is generally well-organized, the authors need to respond to the following points.
1. The sampling methods and the limitations need to be discussed more.
The sampling methods are unclear in the current form. The authors need to answer the following questions: a) How did you ensure the representativeness of the sample? In other words, how do you avoid sampling bias, given that the sampling methods are recruitment by graduate students? c) If there is significant sampling bias, how do you think that the bias affects the results? I know the authors mentioned these points in the discussion section, but the authors need to discuss further how the readers should interpret the results, given the sampling bias.
2. The authors should mention more about limitations based on missing variables.
The contact with nature may be affected by their residential areas and occupations. If the sample lives in the countryside, for example, they have more chances to contact with nature. Also, suppose that their occupations are like a farmer, fisherman, or woodcutter; their opportunity to contact nature and its impact is inevitably greater than office workers, especially if they chose their residential areas and occupations based on their preference.
3. Some sentences are too long and difficult to read.
I have found some very long sentences that last three lines or more. They may serve as an obstacle to the readers. The authors should work more on readability.
4. This is a minor comment, but the first paragraph of the introduction can be improved.
I have no comments on this.
Reviewer 3 Report
I would like to strongly encourage authors to reformulate their manuscript with the changes made in this document.

Academic english language should be improved.
Reviewer 4 Report
Dear Authors,
The submitted manuscript titled „Contact with nature in social deprivation during COVID-19: The positive impact on anxiety” contains the interesting results. The manuscript is generally well-written. However, I have found some imperfections, which-in my opinion-should be improwed or at least clarified before an eventual publication. I have listed them below:
1. Material and methods.
· Lines 215-220. I think, that structure of respondents (based on sociodemographic information) might be presented in Results section. I suggest to present it on graphs.
· In my opinion the procedure of investigations should be more detaiedly desrcibed. There is lack of information when the investigations were conducted.
a. There is lack of justification why near-graduate students were chosen to ivestigations.
b. The questionaire should be presented in supplementary material.
c. The results were analysed statisticaly, but there is lack of information about applied tests and/or procedure of analyses. Therefore, please add the suitable information f.ex. in subchapter Statistical analyses.
d. In my opinion subchapter 2.3 contains to much shortcommings and therefore it is difficult to understand and follow.
2. Results.
The text describing results is overloaded with numbers.
Reviewer 5 Report
The present work is aimed at studying the mechanism through which personal disposition towards nature and contact with nature may promote well-being, measured as a reduction of anxiety and medicine intake.
In general the article is well written, the introduction provides extensive information about the research topic, being it relevant and of interest. Nevertheless, I have some concerns, particularly related to the discussion, I think the authors should tackle:
Lines 302-303. The authors state “it is likely that those people feeling more connected with nature were more motivated to search for green areas where to spend time”. I agree with the statement, this is generally true, but would also expect that more connected people could feel more frustrated because of the limitations to move and spend time in green areas. Do the authors think the lock-down restriction could affect in a larger degree to people more connected with nature?
Lines 304-306. The authors state that “contact with nature was found to promote a restorative experience (H2), but unexpectedly that was not significantly related to a lower level of anxiety (H3)”. The authors should discuss this finding more extensively and address it from the theories of psychological restoration.
Lines 311-312. The authors state “those individuals feeling less anxious about the pandemic situation showed a lower degree of medicine intake, as expected (H4)”. The authors should consider the possibility that feeling less anxious about the pandemic can be consequence of many factors, different from the connections with nature.
Lines 314-317: The authors state “the negative relationship between age and both Contact with Nature and Anxiety suggest that older individuals may engage less in nature-related activities and experience lower anxiety levels”. The authors state that older individuals would experience lower anxiety levels because they rely on medicine intake to manage their health issues. Did the authors control the medicine intake in older adults? I mean, I would expect this result is valid in older adults that use more medicines. What was the CNS score of older adults? similar or different to the rest of sample?
Regarding “females reporting significantly higher anxiety levels than males” (line 324). Are there differences between males and females in CNS scores? I so, what can be reason?, if not why anxiety is different in males and females?
Recently, Sanchez-Nieto et al. (https://doi.org/10.3390/ijerph20032727) have tested the feasibility of VR to display nature scenarios to Alzheimer’s patients, reporting a reduction of psychological and physiological anxiety in the participants. It would be important authors include in the discussion the possibilities of technologies like VR to solve the difficulties to visit nature scenarios (as done in the mentioned paper), in particular for older people and those suffering pathologies that impede moving to the countryside.
Round 2
Reviewer 5 Report
Thanks for your replies to my comments. I think the manuscript is now acceptable for publication.
Author Response
Dear Reviewer,
Thank you for taking the time to review our manuscript and for your feedback. We appreciate your thorough evaluation of our work and the constructive comments you provided.
We are pleased to hear that you find the revised manuscript acceptable for publication. We have carefully considered all your suggestions and incorporated appropriate changes to address the concerns raised during the review process. We believe these revisions have strengthened the quality and clarity of our research.
Once again, thank you for your time, effort, and valuable feedback. We look forward to the next steps in the publication process.
Sincerely,
The Authors